# Assessment of Intercostal Muscle Near-Infrared Spectroscopy for Estimating Respiratory Compensation Point in Trained Endurance Athletes

**DOI:** 10.3390/sports11110212

**Published:** 2023-11-01

**Authors:** Salvador Romero-Arenas, Carmen Daniela Quero-Calero, Oriol Abellan-Aynes, Luis Andreu-Caravaca, Marta Fernandez-Calero, Pedro Manonelles, Daniel Lopez-Plaza

**Affiliations:** 1Facultad de Deporte, UCAM, Universidad Católica de Murcia, 30107 Murcia, Spain; sromero@ucam.edu (S.R.-A.); cdquero@ucam.edu (C.D.Q.-C.); landreu@ucam.edu (L.A.-C.); 2International Chair of Sport Medicine, UCAM, Universidad Católica de Murcia, 30107 Murcia, Spain; mifernandez2@ucam.edu (M.F.-C.); pmanonelles@ucam.edu (P.M.); dlplaza@ucam.edu (D.L.-P.); 3Sports Physiology Department, Faculty of Heatlh Sciences, Catholic Univeristy of Murcia, 30107 Murcia, Spain; 4Faculty of Physiotherapy, Podiatry and Occupational Therapy, UCAM Universidad Católica de Murcia, 30107 Murcia, Spain

**Keywords:** endurance exercise, breakpoint, trail runners, oxygen saturation, training prescription

## Abstract

This study aimed to assess the reliability and validity of estimating the respiratory compensation point (RCP) in trained endurance athletes by analyzing intercostal muscles’ NIRS-derived tissue oxygenation dynamics. Seventeen experienced trail runners underwent an incremental treadmill protocol on two separate occasions, with a 7-day gap between assessments. Gas exchange and muscle oxygenation data were collected, and the oxygen saturation breakpoint (SmO_2_BP) measured in the intercostal muscles was compared to the RCP, which was identified by the increase in the VE/$\dot{V}$CO_2_ slope and the point at which the PetCO_2_ started to decrease. No statistically significant differences were observed between the two methods for any of the variables analyzed. Bland–Altman analysis showed significant agreement between the NIRS and gas analyzer methods for speed (*r* = 0.96, *p* < 0.05), HR (*r* = 0.98, *p* < 0.05), $\dot{V}$O_2_ relative to body mass (*r* = 0.99, *p* < 0.05), and %SmO_2_ (*r* = 0.98, *p* < 0.05). The interclass correlation coefficient values showed moderate to good reliability (0.60 to 0.86), and test–retest analysis revealed mean differences within the confidence intervals for all variables. These findings suggest that the SmO_2_BP measured using a portable NIRS device in the intercostal muscles is a reliable and valid tool for estimating the RCP for experienced trail runners and might be useful for coaches and athletes to monitor endurance training.

## 1. Introduction

Comprehension of the physiological responses to exercise is fundamental for athletes and coaches as it can prevent injury, enhance performance, and even determine running strategies in endurance sports. In such sports, physiological responses are traditionally assessed by measuring oxygen (O_2_) and carbon dioxide (CO_2_) volumes during exercise. In this sense, gas exchange stress tests are valuable tools for assessing an athlete’s cardio-respiratory fitness and metabolic capacity [1].

Certain ventilatory variables, termed “exercise thresholds”, have been known for decades to provide useful information on changes in muscle metabolism and their systemic effects during incremental exercise [2,3,4]. Among the thresholds, the “estimated lactate threshold” and respiratory compensation point (RCP) are the two most commonly recognized. These thresholds are relevant to exercise performance and metabolic capacity, making them an important area of research in exercise physiology [5]. However, measurements of ventilatory and pulmonary gas exchange parameters do not provide specific information on the muscles used during exercise, which limits the comprehensive understanding of systemic changes in the body. To obtain a more complete picture, it is necessary to examine the specific muscles used.

A promising method to obtain this additional information is near-infrared spectroscopy (NIRS), a tool that allows measurement of muscle oxygen saturation (SmO_2_) in targeted muscles [6,7,8,9]. These devices allow for the capture of optical changes at varying wavelengths to quantify SmO_2_ levels in a simple and non-invasive way. NIRS has the potential to provide valuable information on muscle oxygenation during exercise and can help identify limitations that are not observable using traditional pulmonary gas exchange measurements [10]. In sports science, SmO_2_ in locomotor muscles has been extensively investigated, especially in the context of prescribing work intensities in training programs [9,11,12,13,14,15]. However, there has been a notable lack of attention to respiratory muscles, whose SmO_2_ may be conditioned by the demands imposed by different sports disciplines and characteristic ventilatory patterns [13]. The lack of studies on respiratory muscle oxygenation is mainly due to inherent challenges, such as the complex anatomical structure and variability in muscle recruitment. Although deoxygenation has been detected as exercise intensity increases, the current literature lacks studies reporting the degree of concordance of values between ventilatory variables (i.e., RCP), and respiratory muscle SmO_2_. This gap in the literature is of particular concern because of the potential impact of respiratory limitations on endurance exercise performance. Thus, the absence of focused research on SmO_2_ levels in intercostal muscles during physical activity raises a key question: to what extent can SmO_2_ in these muscles serve as a reliable indicator of the RCP in trained endurance athletes?

Therefore, the problem addressed in our study is twofold. First, we sought to assess the reliability of intercostal muscle SmO_2_ measurements during a maximal incremental treadmill test in experienced trail runners using a test–retest design. Second, we aimed to elucidate the relationship between the intercostal muscle oxygen saturation cutoff point (SmO_2_BP) and the RCP. We hypothesized that intercostal muscle SmO_2_ measurements will serve as a valid and reliable tool for estimating the RCP in trained endurance athletes. Resolving these questions will contribute to a more detailed understanding of the role of respiratory mechanics as a potential limiting factor for endurance performance. It is anticipated that the results of this research will not only advance academic discourse but also inform innovative training techniques focused on respiratory muscle development. Therefore, athletes and coaches could gain useful information on how breathing patterns and muscle oxygenation interact to influence athletic performance, providing them with the necessary tools to optimize their results.

## 2. Materials and Methods

### 2.1. Participants

This study included seventeen well-trained male trail runners (age: 32.3 ± 5.42 years; height: 176.2 ± 5.38 cm; body mass: 71.5 ± 8.61 kg) with 9.6 ± 4.56 years of experience, who participated in regular training sessions of 9.0 ± 3.37 h per week and had no history of injury in the previous six months. The sample was recruited by convenience. Exclusion criteria were (a) no injury six months prior to the trials and/or during the trials; (b) a training frequency of less than five times per week, including running or endurance training; and (c) less than two years of trail running experience. All subjects who participated in the study gave written informed consent to the experimental procedures, which were approved by the Ethics Committee of the Catholic University of Murcia of the UCAM (no.: CE022210). Participants were informed about the possible risks and benefits of the study, as well as the possibility of withdrawing from the research at any time without giving a reason or causing any harm or penalty to themselves. The research was conducted in accordance with the most recent Declaration of Helsinki.

### 2.2. Design

In this study, each participant completed the experimental protocol on two occasions, with a seven-day interval between tests, and under identical laboratory environmental conditions. The same investigator conducted both tests and retests to reduce potential sources of intra- and inter-individual variability, and participants were tested at the same time of day during each session. In order to ensure consistency across tests, participants were required to abstain from consuming caffeine or food for at least three hours prior to testing, as well as refraining from engaging in intense physical exercise for a period of 24 h prior to testing. Participants maintained their regular diet, training regimen, and sleep patterns during the seven-day period between tests. Finally, to prevent any potential performance bias, participants were not informed of their test results until the conclusion of the study.

### 2.3. Procedures

#### 2.3.1. Cardiopulmonary Exercise Test

The incremental exercise test consisted of a three-minute warm-up period, during which participants walked at a speed of 4.5 km·h^−1^. Following the warm-up period, the speed was increased to six km·h^−1^, with subsequent increments of one km·h^−1^ every 45 s until the point of exhaustion (i.e., the point at which a person can no longer maintain their exercise intensity or workload due to fatigue or inability to continue). Throughout the test, heart rate and respiratory variables, including respiratory exchange ratio (RER) and $\dot{V}$O_2_, were continuously monitored. The breath-by-breath method was used to obtain ventilation data during the incremental exercise testing, which was subsequently used to evaluate the individual maximal oxygen uptake capacity. Prior to each test, the volume transducer and gas analyzer VyntusTM CPX (Vyaire medical, INC, Hoechberg, Germany) was calibrated in accordance with the manufacturer’s instructions.

#### 2.3.2. Measurement of SmO_2_

The SmO_2_ levels of the intercostal muscles were measured using a non-invasive NIRS instrument (HUMON HEX; Dynometrics Inc., Cambridge, MA, USA). This device consists of six slots that emit LED radiation at two wavelengths (i.e., 760–840 nm), which is used to calculate hemoglobin content (g/dL) and saturation (%) levels. The NIRS sensor was placed on the seventh intercostal space of the anterior axillary line of the right hemithorax, in accordance with established research protocols [11,16]. To secure the device and prevent movement during the test, an adhesive tape and black neoprene sports strapping were used. The sensor’s location was marked with a marker pen to monitor any movement during testing. The device was calibrated following the manufacturer’s guidelines. During testing, the Humon Hex was connected via Bluetooth^®®^ to an Android tablet (Samsung Galaxy Tab A7 lite; Samsung Electronics, Seoul, Republic of Korea), and the MoxZones App (MoxZones SL, Madrid, Spain) was used to display and store the data in real time at an output frequency of 1 Hz.

#### 2.3.3. Skinfold

The intercostal skinfold thickness was measured using a Harpenden skinfold caliper (British Indicators, London, UK) and divided by two [17]. Higher levels of adipose tissue at the measurement site can make it challenging to measure tissue oxygen saturation accurately. The subcutaneous adipose tissue thickness calculated for this study was relatively low, at 3.5 ± 2.06 mm, according to a previous study by Ferrari et al. in 2004 [18].

#### 2.3.4. Data Analysis

The observed variables from the ergospirometry and NIRS devices were manually synchronized by two operators, and the ventilatory and SmO_2_ variables were averaged every 5 s. $\dot{V}$O_2_max was calculated as the average of $\dot{V}$O_2_ values during the last 30 s of the incremental test, and the recommendations by Keir et al. [1] were followed to obtain the RCP. The criteria for achieving the RCP were a non-linear increase in the VE/$\dot{V}$CO_2_ slope accompanied by a second abrupt increase in VE/$\dot{V}$O_2_ with increasing exercise intensity [19], as well as a drop in PetCO_2_.

The SmO_2_BP was determined using visual identification of an SmO_2_ decrease of more than 15% and a subsequent continuous fall [20]. The values for the SmO_2_BP and RCP cutoff points were identified independently for each individual, and the data were aligned in time with the start of exercise at time “zero” [15]. Two experienced researchers verified the values of both measurement methods, resolving conflicts by consensus [7,15].

#### 2.3.5. Statistical Analysis

All statistical analyses were performed using SPSS v24.0 (SPSS Inc., Chicago, IL, USA). The homogeneity of variance and the normal distribution of the sample were examined using Levene’s test and the Kolmogorov–Smirnov test, respectively. To compare the values of each variable at the RCP and the SmO_2_BP, a paired-sample t-test was used and the significance level was set at *p* < 0.05.

A Bland–Altman analysis [21] was conducted to determine the agreement between the SmO_2_BP and the RCP in speed, HR, $\dot{V}$O_2_, and %SmO_2_. The systematic bias and standard deviations of the data were obtained from the correlation between absolute differences and mean values. For the replicability analysis, the intraclass correlation coefficient (ICC) as well as the limits of agreement were calculated. According to Koo and Li (2016), ICC values were identified as low (R < 0.5), moderate (0.5 < R < 0.75), or high (R > 0.9). To examine whether results were different between the test and retest, the mean differences for each variable and their confidence intervals were investigated. When the value of the difference was within the confidence interval, no differences between the test and retest time points were considered.

## 3. Results

The results of the comparisons between the points of the RCP and the SmO_2_BP are summarized in Table 1. The speed measured at the RCP and the SmO_2_BP was not significantly different (*p* > 0.05) between the first and second tests. There were no significant differences (*p* > 0.05) in $\dot{V}$O_2_ between the time of the RCP and the SmO_2_BP in either test. The percentage of $\dot{V}$O_2_max was not significantly different (*p* > 0.05) between the RCP and the SmO_2_BP in either the first or second test. Furthermore, there were no significant differences (*p* > 0.05) in absolute HR value and %HRmax between the RCP and the SmO_2_BP in either the first or second test. RER and SmO_2_ values at the point of the RCP and the SmO_2_BP were not significantly different (*p* > 0.05) in any of the tests performed.

The Bland–Altman analysis of agreement between the SmO_2_BP and the RCP is presented in Figure 1. Mean differences of −0.02 ± 0.62 km·h^−1^ (limits of agreement: lower = −1.23; upper = 1.19 km·h^−1^) and −0.65 ± 2.62 bpm (limits of agreement: lower = −5.78; upper = 4.49 bpm) were observed relative to speed and HR, respectively. In addition, $\dot{V}$O_2_ relative to body mass revealed mean differences of −0.21 ± 1.01 mL·kg·min^−1^ (limits of agreement: lower = −2.20; upper = 1.77 mL·kg·min^−1^) whereas for %SmO_2_ mean values of 0.33 ± 2.59 (limits of agreement: lower = −4.74; upper = 5.40%) were identified. Additionally, strong and significant correlations were identified between instruments in speed (*r* = 0.96; *p* < 0.05), HR (*r* = 0.98; *p* < 0.05), $\dot{V}$O_2_ relative to body mass (*r* = 0.99; *p* < 0.05), and %SmO_2_ (*r* = 0.98; *p* < 0.05).

Table 2 presents the reliability data of the variables measured at the SmO_2_BP. ICC values ranged from 0.60 to 0.86, with SmO_2_ showing the lowest ICC and speed showing the highest ICC. The mean differences between test and retest are shown in the second part of the table. The mean difference for speed was the lowest, while SmO_2_ had the highest mean difference. All mean differences were within the confidence intervals for each variable.

## 4. Discussion

The present study’s results corroborate the hypothesis that a portable NIRS device (Humon HEX; Dynometrics Inc., Cambridge, MA, USA) can provide valid and reliable measurements to estimate the RCP in intercostal muscles during an incremental treadmill test in experienced trail runners. Specifically, significant associations were found between the SmO_2_BP assessed using the Humon HEX monitor and the RCP measured using a gas analyzer across all variables, including speed, heart rate, $\dot{V}$O_2_ relative to body mass, and %SmO_2_. Additionally, the protocol’s reproducibility and variability analysis revealed significant reliability based on ICC values. As the first investigation to test the Humon HEX device on intercostal muscles, these findings could represent a benchmark on the ability of a valid and reliable method to determine the RCP with a portable and relatively inexpensive device (i.e., less than USD 400).

According to the Bland–Altman analysis, very strong correlations were identified between the SmO_2_BP and the RCP (*r* > 0.95). The average discrepancy between the methods remained within the confidence intervals relative to speed, HR, $\dot{V}$O_2_ relative to body mass, and %SmO_2_. Furthermore, the limits of the agreement seem to exhibit narrow ranges compared to the values of each variable examined. Thus, these findings suggest that the two methods used are essentially equivalent (Figure 1). These findings give this assessment the ability to calculate the workload of the respiratory muscles in typical physical training contexts, such as the respiratory compensation point. In this way, we are provided with a new instrument for the accurate determination of exercise intensity. Prior validation studies using near-infrared spectroscopy in the estimation of physiological parameters identified similar or lower relationships between methods. Rodrigo-Carranza et al. (2019) [15] reported moderate to strong correlations (*r* = 0.68–0.86) between oxygen saturation in the vastus lateralis and ventilatory threshold’s two breakpoints relative to %$\dot{V}$O_2_max, absolute speed, HR, and $\dot{V}$O_2_. Similarly, in the analysis of intercostal muscles’ oxygenation, significant associations were observed not only between $\dot{V}$O_2_ and SmO_2_ [11] but also between ventilatory variables and SmO_2_ [22]. Taking all this into consideration and according to the current investigation results, identification of the SmO_2_BP in the intercostal muscles is a valid way to estimate the RCP during incremental treadmill exercise.

Based on the concept of replicability as outlined by Koo et al. [23], our results indicate that portable NIRS technology is a dependable alternative for assessing performance in endurance sports. Specifically, it can estimate the respiratory compensation point (RCP) in intercostal muscles. In the follow-up tests we conducted, all the measured variables showed moderate-to-high intraclass correlation coefficient (ICC) values, which suggests a high level of reliability. This observation was consistent with Contreras et al.’s [11] results, which also reported moderate-to-high and high ICC values in the same muscle group using other NIRS devices. Therefore, the practicality and usefulness of this technology for estimating sports performance parameters in athletes of any level is well-established. Indeed, NIRS-based SmO_2_ estimation has demonstrated high replicability in other muscle groups such as the vastus lateralis [15,24], the gastrocnemius [24], and forearm muscles [25]. However, it is worth noting that the behavior of SmO_2_ in different muscle groups may exhibit slight differences, and further investigation is needed to determine how other factors, such as fitness level or external influences, may affect the relationship between the RCP and the SmO_2_BP and to conclude their interdependence [26].

The use of portable devices such as the Humon HEX for the accurate estimation of physiological variables such as the RCP is a significant advance in the monitoring of sports training. In this way, ventilatory thresholds can be established in real-life settings outside the laboratory without the need for more expensive equipment such as portable lung gas analyzers. Furthermore, the validity and reliability of NIRS devices have been corroborated not only in activities such as running but also in cycling [12,15,26]. Practically, this makes it possible to perform training control tests under real training and competition conditions, as opposed to laboratory testing. Future research could replicate such protocols under field conditions, rather than in the laboratory, to validate the use of NIRS in real training scenarios, considering the full range of external factors. In addition, collecting data with a larger number of points could improve the quality of the work by allowing the observation of potential temporal changes in data analysis.

In terms of the limitations of this study, the sample size is the main limitation. Nevertheless, recruiting a large cohort of experienced runners is a challenge that must be considered. Another potential limitation is the quality of the near-infrared signal due to factors such as adipose tissue thickness. In this regard, the literature reports that thicker adipose tissue affects the penetration of light from the source and thus the recording of data obtained by devices that use NIRS [27]. In this case, subcutaneous adipose tissue was controlled and was relatively low (i.e., 3.5 ± 2.06 mm). All measurements were taken by an experienced anthropometrist following the protocol described in the methods section and at the same time of day.

## 5. Conclusions

The results of the present study underscore the efficacy of portable NIRS technology, particularly the Humon HEX device, in estimating the RCP during incremental treadmill exercises among experienced trail runners. These findings establish that the method is both valid and reliable for assessing the RCP in the intercostal muscles. By revealing a strong association between the SmO_2_BP and the RCP, this study suggests that NIRS is a reliable tool for identifying crucial physiological markers. As such, NIRS technology holds significant potential for the real-time monitoring of sports training outside the laboratory, allowing for immediate adjustments to training and competition strategies. It is recommended as a valuable resource for both coaches and advanced runners to personalize and oversee training programs from an exercise physiology perspective.

## 6. Practical Applications

The present study demonstrates the usefulness and efficacy of the portable near-infrared spectroscopy device Humon HEX in the reliable and valid assessment of intercostal muscle CPR during incremental treadmill testing in experienced trail runners. These findings have immediate and long-term practical applications for both coaches and athletes. First, the device allows real-time monitoring of CPR, eliminating the need for laboratory testing and facilitating adjustments in training and competition strategies. Second, the Humon HEX offers a more economically and logistically accessible alternative to laboratory gas analyzers. This is particularly relevant for field tests and regular performance monitoring. Furthermore, the methodological soundness of this study supports the applicability of NIRS devices in other endurance sports such as cycling, thus extending their usefulness in various athletic contexts. Finally, these results can serve as a basis for future research seeking to validate NIRS technology in real training situations considering external and environmental factors.

## Figures and Tables

**Figure 1 sports-11-00212-f001:**
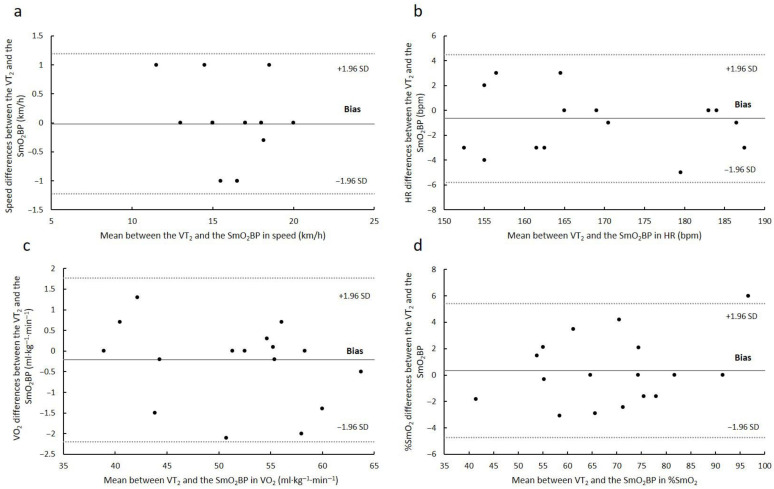
Bland–Altman plots for the near-infrared spectroscopy device at the RCP in (**a**) speed, (**b**) heart Rate, (**c**) $\dot{V}$O_2_ relative to body mass; and (**d**) %SmO_2_. The solid line represents the absolute average difference between instruments while the upper and lower dotted lines indicate ± 1.96 standard deviations (SD).

**Table 1 sports-11-00212-t001:** Values of speed, heart rate, oxygen uptake, muscle oxygen saturation, and respiratory exchange rate at the respiratory compensation point and muscle-oxygen-saturation breakpoint.

Variable	Test (*n* = 17)					Retest (*n* = 17)				
	RCP		SmO_2_BP		*p*	RCP		SmO_2_BP		*p*
	M	SD	M	SD		M	SD	M	SD	
Speed (km·h^−1^)	16.1	2.09	16.1	2.22	0.908	15.2	1.78	15.4	1.90	0.269
$\dot{V}$O_2_ (ml·kg^−1^·min^−1^)	52.5	8.36	52.7	8.42	0.401	49.8	5.97	49.8	7.07	0.930
Percentage of $\dot{V}$O_2_max (%)	88.5	4.10	88.8	4.08	0.447	88.4	5.37	88.3	5.32	0.891
HR (bpm)	170.8	13.63	171.5	13.33	0.324	167.1	12.77	168.2	13.45	0.263
Percentage of HRmax (%)	94.1	3.08	94.4	3.00	0.311	92.8	2.90	93.4	3.13	0.282
SmO_2_ (%)	68.9	14.57	68.6	13.81	0.607	62.3	11.22	63.0	11.12	0.098
RER	1.01	0.05	1.01	0.05	0.590	1.02	0.06	1.02	0.06	0.906

$\dot{V}$O_2_: oxygen uptake; HR: heart rate; SmO_2_: muscle oxygen saturation; RER: respiratory exchange ratio; BP: breakpoint; RCP: respiratory compensation point.

**Table 2 sports-11-00212-t002:** Intraclass correlation and mean differences between test and retest for speed, $\dot{V}$O_2_, HR, and SmO_2_ at the SmO_2_BP.

	ICC		Mean Differences	
Variable	R	95% CI	Mean Differences	95% CI
Speed (km·h^−1^)	0.859	0.65–0.94	−0.78	−1.34–(−0.22)
$\dot{V}$O_2_ (mL·kg·s^−1^)	0.793	0.52–0.92	−2.68	−5.22–(−0.13)
HR (bpm)	0.823	0.57–0.93	−3.29	−7.39–0.81
SmO_2_ (%)	0.600	0.18–0.83	−5.60	−11.36–0.17

$\dot{V}$O_2_: oxygen uptake; HR: heart rate; SmO_2_: muscle oxygen saturation.

## Data Availability

The data that support the findings of this study are available from the corresponding author upon reasonable request.

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
