# Peer review of "Assessment of Intercostal Muscle Near-Infrared Spectroscopy for Estimating Respiratory Compensation Point in Trained Endurance Athletes"

_sports, 2023, doi:10.3390/sports11110212_

Round 1
Reviewer 1 Report
Comments and Suggestions for Authors
Thank you for the opportunity to review the current review paper. The authors had decided to examine an important topic in exercise physiology and should be commented for their work. However, there are a few points that need to be addressed. Here are my comments, line by line.
Line 21: B-A plot is a measure of agreement, not correlation. Please amend.
Line 33/34: Typo here, and throughout the text (O2 should be given as O2, and the same goes for CO2);
Line 69: Please develop a study hypothesis here.
Line 99: Typo, please amend.
Table 2: Typo here as well, mL kg s -1, should be give as mL kg min -1, right ?
VO2 should be presented as V̇O2, as this is the correct way to indicate flow, here and throughout the MS.
Line 207: Too strong statement in my opinion. Here, you have only n=17, so please re-write this sentence.
Line 22: reliable SURROGATE measure
Line 242: You should imply here that data collection with more data points should add to the quality of work as we would see changes over time in the data analysis.
Comments on the Quality of English LanguageThank you for the opportunity to review the current review paper. The authors had decided to examine an important topic in exercise physiology and should be commented for their work. However, there are a few points that need to be addressed. Here are my comments, line by line.
Line 21: B-A plot is a measure of agreement, not correlation. Please amend.
Line 33/34: Typo here, and throughout the text (O2 should be given as O2, and the same goes for CO2);
Line 69: Please develop a study hypothesis here.
Line 99: Typo, please amend.
Table 2: Typo here as well, mL kg s -1, should be give as mL kg min -1, right ?
VO2 should be presented as V̇O2, as this is the correct way to indicate flow, here and throughout the MS.
Line 207: Too strong statement in my opinion. Here, you have only n=17, so please re-write this sentence.
Line 22: reliable SURROGATE measure
Line 242: You should imply here that data collection with more data points should add to the quality of work as we would see changes over time in the data analysis.
Author Response
Thank you for the opportunity to review the current review paper. The authors had decided to examine an important topic in exercise physiology and should be commented for their work. However, there are a few points that need to be addressed. Here are my comments, line by line
First of all, thank you for the time taken to review the paper and the useful suggestions and comments. The responses are shown individually, in blue:
Line 21: B-A plot is a measure of agreement, not correlation. Please amend.
It has been corrected
Line 33/34: Typo here, and throughout the text (O2 should be given as O2, and the same goes for CO2);
It has been corrected
Line 69: Please develop a study hypothesis here.
The following has been included:
“We hypothesized that intercostal muscle SmO2 measurements will serve as a valid and reliable tool for estimating RCP in trained endurance athletes.”
Line 99: Typo, please amend.
Corrected
Table 2: Typo here as well, mL kg s -1, should be give as mL kg min -1, right ?
You are right. It has been corrected
VO2 should be presented as V̇O2, as this is the correct way to indicate flow, here and throughout the MS.
It was this way, but in Palynotype font the dot was out of place for some reason. Another font has been included in the V to amend it in the whole MS
Line 207: Too strong statement in my opinion. Here, you have only n=17, so please re-write this sentence.
It has been modified as follows:
“As the first investigation to test the Humon HEX device on intercostal muscles, these findings could represent a benchmark on the ability of a valid and reliable method to determine RCP with a portable and relatively inexpensive device (i.e., less than 400 USD).”
Line 22: reliable SURROGATE measure
Included
Line 242: You should imply here that data collection with more data points should add to the quality of work as we would see changes over time in the data analysis.
The following has been included:
“In addition, collecting data with a larger number of points could improve the quality of the work by allowing the observation of potential temporal changes in data analysis.”
Reviewer 2 Report
Comments and Suggestions for Authors
Manuscript title:
2651644-Assessment of intercostal muscle near-infrared spectroscopy for estimating respiratory compensation point in trained endurance athletes
Títle |
|
Is it understandable and concise? |
( x ) Yes ( ) Not |
Reflects the content? |
( x ) Yes ( ) Not |
Abstract |
|
It includes: objectives, methodology, key findings and conclusions? |
( ) Yes ( x ) Not |
Introduccion |
|
The investigation was carried out in a suitable theoretical structure? |
( ) Yes ( x ) Not |
Clear leaves the questions you want to answer and objectives of the work? |
( ) Yes ( x ) Not |
The cited references are current and relevant? |
( ) Yes ( x ) Not |
Methods |
|
The methods presented are appropriate to achieve the proposed objectives? |
( ) Yes ( x ) Not |
The selection and composition of the sample are adequately described? |
( ) Yes ( x ) Not |
The data collection process and the tools used are described clearly? |
( ) Yes ( x ) Not |
The statistical analysis and the research design appropriate? |
( ) Yes ( x ) Not |
Results |
|
The presentation of the results clear? |
( ) Yes ( x ) Not |
The main results are highlighted without the inclusion of interpretation and comparisons? |
( ) Yes ( x ) Not |
The results evaluate the proposed objectives? |
( ) Yes ( x ) Not |
Tables and figures are properly numbered, labeled and explained? |
( ) Yes ( x ) Not |
Discussion and Conclusion |
|
The results are discussed based on the literature? |
( ) Yes ( x ) Not |
Author's interpretations show the safety and soundness? |
( ) Yes ( x ) Not |
The limitations of the work are presented? |
( x ) Yes ( ) Not |
The conclusions of the study are presented? |
( ) Yes ( x ) Not |
The conclusions respond to the objectives? |
( ) Yes ( x ) Not |
General comments:
Title
Are presented satisfactorily.
Abstract
It is written in a structured way, however, the methodology starts with background, but this was not presented but the objective of the study. Please correct. Initially place the background, before the objectives.
In addition to the statistical values, it would be recommended to include the main absolute values found in the study.
In the conclusions, it would be recommended to include the practical applications of the findings.
Please check whether the descriptors are listed as health science descriptors.
Introduction
The introduction does not go from the general to the specific. Methodological aspects are presented, including the names of instruments such as NIRS. This should only be presented in the methodology.
Statements were made that the parameters that will be studied and not the equipment should be limited.
The problem is not well defined. Just mention that the NIRS is a new piece of equipment that tends to evaluate oxygenation at a muscular level, it is not enough. It would be feasible to present studies for and against more traditional respiratory parameters and the parameters, but not the device, that will be the objects of study. Thus, the studies for and against will be demonstrated and the scientific gap will be presented.
The introduction is not starting from general to specific.
It should initially present a more general approach, gradually address the problem (gap), and then present the objective.
After the problem was presented, the objectives of the study would be presented.
After presenting the objectives, the hypotheses to be answered by the study will be presented.
Methods
It should present more clearly the design of the study. A CONSORT or timeline, should be presented in order to get a better view of the study design. I suggest reversing the topics and placing the drawing before the participants.
The sample should be better explained with the number of subjects presented initially and then present the inclusion and exclusion criteria. No statistical analyses or even the use of any programs were presented, or even other studies that could justify the sample size. Please add this to the participant's topic.
It would be important, along with the procedures, to mention the instruments more broadly, including model, and in parentheses manufacturers, city, state if any, and country of manufacture. Furthermore, if there is any study that validated the instrument, it must be presented, or even another study that used the instrument. In the procedures, it would be recommended to present references that justify the use of the equipment and how it was used, as well as possible cutoff points, if any.
Statistical treatment should be better detailed in order to better follow what has been done. The quote from Koo and Li was made, however, in violation of the magazine's rules. It would be interesting to use, consult, and reference an author who would be the reference in terms of the proposed analyses. Suggest you consult Cohen J. Statistical Power Analysis for the Behavioral Sciences (2nd ed.), New Jersey: Lawrence Erlbaum Associates, 1988
Results
In the table, a test and retest were performed, however, the evaluations were carried out only in the test and then retest. The test was not referenced in the methodology. Another question, would this be the most correct analysis, or would it be feasible to analyze the test and retest together? Please explain. For both variables, other than p must be presented for the statistics to be considered correct. Please consult Cohen (1988).
Before Bland Altman's presentation, wouldn't other evaluations be opportune so that the test can then confirm the results presented?
With regard to Table 2, after evaluating the aforementioned, the table should probably be relocated or completed.
Discussion
It should reaffirm the objectives and start discussing the results in the chronological order that appears in the item results. This was presented.
However, due to what is presented in the methodology and in the presentation of the results, the discussion must be further in-depth in order to justify the findings.
Conclusion
Are presented satisfactorily. However, practical applications of the findings must be presented.
References
Of the 26 references presented, six are current and 20 have been published for more than five years. Please format references in accordance with the journal's standards and update the theoretical framework.
Overview
The manuscript presented addresses a relevant research topic.
It would be advisable to do a general review.
Specific comments and suggestions:
Outcome evaluation
Author Response
First of all, thank you for the time taken to review the paper and the useful suggestions and comments. The responses are shown individually, in blue:
Title
Are presented satisfactorily.
Abstract
It is written in a structured way; however, the methodology starts with background, but this was not presented but the objective of the study. Please correct. Initially place the background, before the objectives.
Thank you for your comments. The abstract has been corrected taking into account your considerations and the journal's standards.
“The abstract should be a single paragraph and should follow the style of structured abstracts, but without headings: 1) Background: Place the question addressed in a broad context and highlight the purpose of the stud….”
In addition to the statistical values, it would be recommended to include the main absolute values found in the study.
The instructions indicate that the abstract should have a maximum of 200 words. Including the main absolute values would increase the number of words over the maximum. The abstract has slightly over 200 words already (including practical applications as the reviewer suggested below)
In the conclusions, it would be recommended to include the practical applications of the findings.
They have been mentioned briefly at the end of the abstract due to number of word restriction
Please check whether the descriptors are listed as health science descriptors.
Most of the descriptors are listed in Health Science. However, a few belong to Sport Science since this research might be also considered of that field.
Introduction
The introduction does not go from the general to the specific. Methodological aspects are presented, including the names of instruments such as NIRS. This should only be presented in the methodology. Statements were made that the parameters that will be studied and not the equipment should be limited.
Thank you for your guidance. We have revised the introduction in line with your suggestions. Additionally, we have removed all content related to the methodological aspects of the study. However, as this is an instrument validation study, we believe it is essential to retain the NIRS technology as the foundational basis of the research.
The problem is not well defined. Just mention that the NIRS is a new piece of equipment that tends to evaluate oxygenation at a muscular level, it is not enough. It would be feasible to present studies for and against more traditional respiratory parameters and the parameters, but not the device, that will be the objects of study. Thus, the studies for and against will be demonstrated and the scientific gap will be presented.
Thank you for your feedback. We have revised the concluding section of the introduction to articulate the research question more explicitly.
The introduction is not starting from general to specific. It should initially present a more general approach, gradually address the problem (gap), and then present the objective. After the problem was presented, the objectives of the study would be presented. After presenting the objectives, the hypotheses to be answered by the study will be presented.
Thank you for your comments. We have made the necessary corrections, taking all of your feedback into account.
Methods
It should present more clearly the design of the study. A CONSORT or timeline, should be presented in order to get a better view of the study design. I suggest reversing the topics and placing the drawing before the participants.
Thank you for noticing. In some other longitudinal studies conducted by us we have presented this kind of timelines; however the subjects of this study only attended the lab once. Thus, the figure would just show the same test twice as this study focused on validity and reliability.
The sample should be better explained with the number of subjects presented initially and then present the inclusion and exclusion criteria. No statistical analyses or even the use of any programs were presented, or even other studies that could justify the sample size. Please add this to the participant's topic.
Thank you for noticing, we have added more information relevant to the description of the sample addressing the issues detected by reviewer #2.
No sample size were calculated due to a convenience sample recruitment. As most of studies conducted on this population include approximately between 7 and 22 subjects, this study was per-formed once 17 trail runners were obtained.
It would be important, along with the procedures, to mention the instruments more broadly, including model, and in parentheses manufacturers, city, state if any, and country of manufacture. Furthermore, if there is any study that validated the instrument, it must be presented, or even another study that used the instrument. In the procedures, it would be recommended to present references that justify the use of the equipment and how it was used, as well as possible cutoff points, if any.
Models, manufacturers, city and country are presented for the instruments used (VyntusTM CPX (Vyaire medical, INC, Hoechberg, Germany); NIRS instrument (HUMON HEX; Dynometrics Inc., Cambridge, MA, USA).)
Validated studies that previously used these devices are presented: The NIRS sensor was placed on the seventh intercostal space of the anterior axillary line of the right hemithorax, in accordance with established research protocols [11,16].
Statistical treatment should be better detailed in order to better follow what has been done. The quote from Koo and Li was made, however, in violation of the magazine's rules. It would be interesting to use, consult, and reference an author who would be the reference in terms of the proposed analyses. Suggest you consult Cohen J. Statistical Power Analysis for the Behavioral Sciences (2nd ed.), New Jersey: Lawrence Erlbaum Associates, 1988
We presented the Koo and Li (2016) for interpretation of ICC as current studies on this topic use this reference and the discussion of the results can be better addressed by using same statistical methods. As well, the reference used is more recent than Cohen’s (1988).
Results
In the table, a test and retest were performed, however, the evaluations were carried out only in the test and then retest. The test was not referenced in the methodology. Another question, would this be the most correct analysis, or would it be feasible to analyze the test and retest together? Please explain. For both variables, other than p must be presented for the statistics to be considered correct. Please consult Cohen (1988).
In the method section, we presented more clearly explanation about the test re-test method. We chose this kind of analysis for a better replication of results done by previous studies on the same topic.
Rodrigo-Carranza, V., González-Mohíno, F., Turner, A. P., Rodriguez-Barbero, S., & González-Ravé, J. M. (2021). Using a portable near-infrared spectroscopy device to estimate the second ventilatory threshold. International Journal of Sports Medicine, 42(10), 905-910.
Before Bland Altman's presentation, wouldn't other evaluations be opportune so that the test can then confirm the results presented? With regard to Table 2, after evaluating the aforementioned, the table should probably be relocated or completed.
What other evaluations should be presented before Bland Atlman’s? We will be happy to present it if reviewer #2 indicates it
Discussion
It should reaffirm the objectives and start discussing the results in the chronological order that appears in the item results. This was presented. However, due to what is presented in the methodology and in the presentation of the results, the discussion must be further in-depth in order to justify the findings.
Thank you for your valuable feedback. We appreciate your recommendation to deepen the discussion section in order to more comprehensively justify the findings. We have revised the discussion to state the objectives of the study more robustly for clarity and consistency.
Conclusion
Are presented satisfactorily. However, practical applications of the findings must be presented.
Thank you for your considerations. We have added a section on practical applications as suggested by the reviewer.
It reads as follows:
- Practical Applications
The present study demonstrates the usefulness and efficacy of the portable near-infrared spectroscopy device, Humon HEX, in the reliable and valid assessment of intercostal muscle CPR during incremental treadmill testing in experienced trail runners. These findings have immediate and long-term practical applications for both coaches and athletes. First, the device allows real-time monitoring of CPR, eliminating the need for laboratory testing, and facilitating adjustments in training and competi-tion strategies. Second, the Humon HEX offers a more economically and logistically accessible alternative to laboratory gas analyzers. This is particularly relevant for field tests and regular performance monitoring. Furthermore, the methodological soundness of this study supports the applicability of NIRS devices in other endurance sports such as cycling, thus extending their usefulness in various athletic contexts. Finally, these results can serve as a basis for future research seeking to validate NIRS technology in real training situations considering external and environmental factors.
References
Of the 26 references presented, six are current and 20 have been published for more than five years. Please format references in accordance with the journal's standards and update the theoretical framework.
Some of the references used are old but we considered the as necessary because they state the basis of the analysis (2005-2010). Some are more than 5 years but still recent i.e. 2012-2017. Nevertheless, there are some old references that are not necessarily important for this study. We have added some more recent references substituting oldest ones.
Overview
The manuscript presented addresses a relevant research topic.
It would be advisable to do a general review.
Thank you for your suggestion. We have carried out the changes asked and we hope that the quality of the manuscript increased accordingly to the changes done.
Round 2
Reviewer 2 Report
Comments and Suggestions for Authors
Diante dos ajustes realizados, considero o manuscrito em condições de ser publicado